# Overview of CircRNAs Roles and Mechanisms in Liver Fibrosis

**DOI:** 10.3390/biom13060940

**Published:** 2023-06-05

**Authors:** Gaiping Wang, Jiahui Tong, Yingle Li, Xianglei Qiu, Anqi Chen, Cuifang Chang, Guoying Yu

**Affiliations:** 1State Key Laboratory of Cell Differentiation and Regulation, Henan Normal University, Xinxiang 453007, China; 041165@htu.edu.cn (G.W.); 2204183035@stu.htu.edu.cn (J.T.); 2104283139@stu.htu.edu.cn (Y.L.); 2204283135@stu.htu.edu.cn (X.Q.); 2104183065@stu.htu.edu.cn (A.C.); 2Henan Center for Outstanding Overseas Scientists of Organ Fibrosis, Henan Normal University, Xinxiang 453007, China; 3Henan International Joint Laboratory of Pulmonary Fibrosis, Henan Normal University, Xinxiang 453007, China; 4Institute of Biomedical Science, Henan Normal University, Xinxiang 453007, China; 5College of Life Science, Henan Normal University, Xinxiang 453007, China

**Keywords:** circRNA, liver fibrosis, ceRNA

## Abstract

Liver fibrosis represents the reversible pathological process with the feature of the over-accumulation of extracellular matrix (ECM) proteins within the liver, which results in the deposition of fibrotic tissues and liver dysfunction. Circular noncoding RNAs (CircRNAs) have the characteristic closed loop structures, which show high resistance to exonuclease RNase, making them far more stable and recalcitrant against degradation. CircRNAs increase target gene levels by playing the role of a microRNA (miRNA) sponge. Further, they combine with proteins or play the role of RNA scaffolds or translate proteins to modulate different biological processes. Recent studies have indicated that CircRNAs play an important role in the occurrence and progression of liver fibrosis and may be the potential diagnostic and prognostic markers for liver fibrosis. This review summarizes the CircRNAs roles and explores their underlying mechanisms, with a special focus on some of the latest research into key CircRNAs related to regulating liver fibrosis. Results in this work may inspire fruitful research directions and applications of CircRNAs in the management of liver fibrosis. Additionally, our findings lay a critical theoretical foundation for applying CircRNAs in diagnosing and treating liver fibrosis.

## 1. Introduction

Various chronic irritations can damage the liver, which thus causes inflammation and necrosis of hepatocytes, activates the hepatic stellate cells, and produces the significant accumulation of extracellular matrix (ECM) that is extremely rich in type I collagen and other collagens [1], eventually resulting in liver fibrosis. Liver fibrosis, a common pathological change in various chronic liver diseases [2], presents as a precursor to liver cirrhosis, and it finally develops into liver failure and hepatocellular carcinoma (HCC) [3]. Thus, it is critical to slow down, or ideally reverse liver fibrosis progression. According to recent articles, many types of CircRNAs are related to liver fibrosis occurrence and development. For instance, CircPWWP2A has been suggested to enhance hepatic stellate cells (HSCs) growth and activation and liver fibrosis by acting as the sponge of miR-203 and miR-223, which then up-regulate Fstl1 and Tlr4, respectively [4]. However, biological functions and molecular mechanisms underlying CircRNAs in hepatic fibrosis (HF), particularly in relation to the activation of HSCs, remain ambiguous and require further investigations. This review summarizes the functionality and regulatory processes of CircRNAs in liver fibrosis and provides the potential diagnostic and therapeutic targets for liver fibrosis, as well as the potential research pathways for related liver diseases. 

## 2. Liver Fibrosis

Pathologically, liver fibrosis refers to the excessive proliferation of connective tissues within the liver induced by different factors, and pathological processes involved in excess diffuse deposition of extracellular matrices within the liver. This represents the final common strategy employed by the liver for self-repair following injury caused by various chronic liver diseases [5], such as chronic viral hepatitis [6], metabolic disorders [7], alcoholic liver disease (ALD), nonalcoholic steatohepatitis (NASH), and autoimmune imbalance [5,8]. Additionally, there are several metabolically related liver diseases [5] such as biliary obstruction [9], for which hepatitis A-E viruses have been recognized as primary initiators [10]. However, importantly, liver fibrosis stands for the reversible wound-healing process, which focuses on maintaining the integrity of organs. Nevertheless, this disease also represents the critical stage toward liver cirrhosis [3], since persistent HF can result in cirrhosis, and even liver failure or hepatocellular carcinoma (HCC) [11]. 

Diverse types of cells and mediators are implicated in HF development, where HSCs have an important effect on its initiation and progression [12], as they are the primary source cells of fibrous matrices and the main effector cells for liver fibrosis [13]. HSCs are situated in perisinusoidal space between sinusoidal endothelial cells and hepatocytes, where quiescent HSCs (qHSCs) are typically the vitamin A reserve [3]. In the case of liver injury, the qHSCs are activated upon cytokine stimulation, such as interleukin (IL)-6 [14,15], IL-17 [16,17,18], and IL-22 [17,18] that are secreted from neighboring cells such as Kupffer cells, hepatocytes, leukocytes, and sinusoidal endothelial cells [14], and these cells can transdifferentiate into the activated myofibroblast-like cells [19,20], fibroblast-like cells, and myofibroblasts. Further, lipopolysaccharide (LPS) and transforming growth factor-beta (TGF-β) are found to induce HSC activation; therefore, up-regulating type I collagen alpha 1 (Col1α1) and alpha-smooth muscle actin (α-SMA). This leads to the excessive accumulation of ECM proteins (including collagen I, collagen III, laminin, and fibronectin) [7], which distorts the liver architecture through the formation of fibrous scars, where hepatocytes are replaced by abundant ECM [21]. Simultaneously, the decreased lipid droplets [20] and vitamin A levels, and the increased rough endoplasmic reticulum and microfilaments in the HSCs eventually lead to the development of liver fibrosis [20]. Therefore, the inhibition or depletion of HSCs has been considered to be the candidate treatment for liver fibrosis [22]. 

## 3. CircRNAs Production and Functional Mechanisms

CircRNAs are noncoding RNAs with high abundances within mammalian cells, which represent the single-stranded circular transcripts that are generated via reverse splicing [23,24]. They possess the covalently-linked head-to-tail closed loop structures, with no 5′–3′ polarity or the polyadenylated tail [25,26]. Due to the closed loop structure, CircRNAs exhibit resistance to RNA exonucleases and are more stable [27,28]. Unlike traditional linear RNAs, CircRNAs show higher abundances in cytoplasm and exosomes of mammalian cells [29], which exhibit high expression in specific tissues, species, and diseases [30]. 

The backsplicing of CircRNA directly competes with canonical splicing events; therefore, only CircRNA production may occasionally induce the decreased cognate linear RNA expression, and/or alterations in linear isoforms present within cells [31]. On the other hand, the splicing mode of pre-mRNA can directly compete with CircRNA biogenesis, thus affecting the CircRNA levels [32]. For example, the alternative splicing factor, quaking (QKI), can regulate circRNA biogenesis via binding sites in introns during epithelial-mesenchymal transition (EMT) [33], and RNA-binding protein 20 (RBM20) is reported to modulate circRNA production from the Titin gene by excluding specific exons from the pre-mRNA [34]. On the other hand, the production of SCD-circRNA 2 is dynamically regulated by RNA-binding protein 3 (RBM3) in HCC cell lines, but it remains unclear how RBM3 affects the alternative splicing of SCD-circRNA 2 [35]. Furthermore, intronic repeat sequences are also factors affecting the biogenesis of CircRNAs in animals [36,37]. It has also been discovered that adenosine deaminase acting on RNAs (ADARs) plays a role as potent regulators of the circular transcriptome in different types of cancer cells, including esophageal, breast, colon, liver, and gastric cancers [38]. Moreover, RNA editing from adenosine-to-inosine (A-to-I) is reduced in heart failure, and CircRNA formation is up-regulated by regulating the RNA stability of ADAR2, which provides further insight into the regulation of CircRNA formation by RNA editing [39]. Additionally, the state of cell division affects the CircRNA levels, such as the higher levels in slowly dividing cells and the lower levels in rapidly dividing cells [40,41,42]. Therefore, CircRNA production is affected by the transcriptional level, post-transcriptional level, and cell proliferation status. 

Collectively, mature CircRNAs have key effects on regulating gene expression in different ways. (1) CircRNAs act as competitive endogenous RNAs (ceRNAs) to regulate the bioavailability of miRNAs [43], such as CDR1as/CIRS-7 [44]. (2) CircRNAs may serve as groups for sequestering dsRNA-binding proteins, which prevent these protein factors from acting elsewhere as sponges or decoys for protein-binding proteins [43]. For instance, CircRNA-SORE can bind to the master oncogenic protein YBX1 in the cytoplasm, which then prevents YBX1 nuclear interaction with the E3 ubiquitin ligase PRP19 and thus blocks the PRP19-mediated YBX1 degradation [45]. (3) CircRNAs are the RNA scaffolds, as certain CircRNA-protein interactions can promote complexes (CircRNPs) formation for regulating gene expression [31]. To take an example, CircRNA SCAR transcript encoded by mitochondrial DNA can inhibit the outputs of mitochondrial reactive oxygen species (ROS) through the formation of complexes with ATP synthase 5B (ATP5B), and the closure of mitochondrial permeability transition pores (mPTPs) [46]. In addition, Wang et al. found that hsa_circ_0074854 promoted the progression of HCC by interacting with human antigen R (HuR, an RNA-binding protein) and inhibiting the exosome-mediated M2 polarization of macrophages [47]. (4) Circular RNAs may serve as protein scaffolds. For example, the formed circ-Foxo3-p21-CDK2 ternary complexes can arrest CDK2 functions and block the cell cycle progression [48] or play a role of critical scaffolds within mitochondria and nuclei [31]. Importantly, CircRNAs always modulate corresponding parental gene transcription through modulating RNA polymerase II [32,49]. (5) CircRNAs translate proteins by serving as a template for protein translation, such as the FBXW7 gene, which is the extensively studied tumor suppressive E3 ligase encoding the new protein (185 amino acids) within human cells [50] (Figure 1). 

With further comprehensive investigations into the expression and functions of CircRNAs, it has become increasingly clear that CircRNAs exert powerful regulatory functions in many diseases, which are thus the candidate markers used to diagnose and treat several diseases [51,52,53].

## 4. Mechanism of CircRNAs in Liver Fibrosis

Many CircRNAs have been widely suggested to be related to promoting or inhibiting the development and progression of liver fibrosis; therefore, they may serve as valuable biomarkers for monitoring liver fibrosis development (Figure 2) [4,21,54,55]. For example, based on experimental data from Zhou et al., CircRNAs expression profiles within liver fibrosis tissues were remarkably altered. Some of the CircRNAs with differential expression are closely related to physiological processes, such as stellate cell activation, macrophage inflammation, and oxidative stress damage. The target genes of CircRNAs are primarily involved in pathways, such as HIPPO, TGF-β1/Smads, vascular endothelial growth factor (VEGF), and RAP1 [56]. Meanwhile, Chen et al. used the CircRNA microarray for investigating CircRNA expression profiles of irradiated HSCs from HCC. According to their results, there were 179 CircRNAs showing up-regulation, whereas 630 displayed down-regulation. Bioinformatics analysis implied that the abnormal expression of CircRNAs was possibly related to cell responses to radiation and the biological process of liver fibrosis [57]. Therefore, elucidating the relationships between the aberrant expression of CircRNAs within HSCs with liver fibrosis is of crucial importance. 

### 4.1. CircRNAs Are Involved in Inhibiting Liver Fibrosis

CircRNAs related to inhibiting liver fibrosis have similar characteristics to their down-regulated expression for this condition, as they always inhibit HSC growth and activation upon their overexpression and suppress fibrosis development. Several CircRNAs (such as CircCREBBP [54], CircFBXW4 [21], CircPSD3 [12], hsa_circ_0070963 [58], hsa_circ_0004018 [22], mmu_circ_34116 [59], CircDIDO1 [60], CircMTO1 [6,61] and mmu_circ_0000623 [62]) are found to suppress liver fibrosis (Table 1). Consequently, such CircRNAs play important roles in liver fibrosis progression and HSC activation, which are the potential therapeutic strategies for liver fibrosis. 

#### 4.1.1. CircRNAs Are Related to Liver Fibrosis through Serving as RNA Scaffolds

##### CircRNA SCAR

After analyzing CircRNA expression patterns in primary liver fibroblasts collected in NASH cirrhosis cases, Zhao et al. uncovered that 15 and 11 CircRNAs displayed up-regulation and down-regulation, separately, within NASH fibroblasts, including four mitochondria-specific CircRNAs. Next, those four mitochondria-specific CircRNAs were introduced into the NASH fibroblasts by synthesizing the mitochondrial-targeted nanoparticles (mito-NP). It was found that only the hsa_circ_0089762 overexpression (steatohepatitis-associated CircRNA ATP5B regulator [SCAR]) significantly reduced the level of cytosolic ROS (cROS). This clearly inhibited the contractility, collagen generation, α-SMA expression, and cytokine secretion in NASH fibroblasts, thus alleviating the fibrotic phenotype [46]. 

The main mechanism involves the direct binding of CircRNA SCAR to ATP5B within the mPTP complex, which closes mPTP by blocking CypD-MPTP interactions and reducing the generation and release of mROS, thereby inhibiting the activation of fibroblasts. In the NASH liver fibroblasts, lipid overload can up-regulate the endoplasmic reticulum stress protein CHOP. This inhibits PGC-1a (peroxisome proliferator-activated receptor gamma, coactivator 1 alpha), then suppresses the transcription of CircRNA SCAR, and eventually promotes the activation of fibroblasts [46]. Researchers have given mito-NP that contains CircRNA SCAR overexpression vectors (or empty vectors) to mice that were fed with a high-fat diet through intravenous administration. According to their observations, the increased CircRNA SCAR significantly inhibits the mPTP opening, cROS release, and collagen contraction of mouse fibroblasts. This obviously improves glucose and insulin tolerance and halts weight gain, while reducing fibrosis and macrophage infiltration in the mouse livers. In clinical practice, CircRNA SCAR is related to steatosis-to-NASH progression [46]; in this regard, a mitochondrial CircRNA SCAR modulates lipid metabolism and may be the anti-NASH therapeutic target. 

#### 4.1.2. CircRNAs Are Related to Liver Fibrosis by Playing a Role of miRNA-Binding Cernas

##### CircCREBBP

As a highly conserved transcription coactivator, CREBBP possesses histone acetyltransferase activities [76]. However, its mutation or deficiency is frequently detected in human cancers, which suggests that it may serve as a tumor suppressor in pathophysiological processes [77,78]. CircCREBBP (hsa_circ_0007673, mmu_circ_0006288) originates from the backsplicing of the CREBBP exons [54] located in 16p13.3 [79]. 

Through conducting high-throughput sequencing, Yang et al. detected CircRNAs expression patterns within the fibrotic liver. As a result, there were 103 CircRNAs showing differential expression, including 18 with up-regulation and 85 with down-regulation within HF tissues [54]. Among them, CircCREBBP exhibited significant downregulation among HF mice. As revealed by the AAV8-induced CircCREBBP overexpression within mice, CircCREBBP significantly suppresses HSCs growth and activation because it blocks the cell cycle, reduces the transdifferentiation of myofibroblasts, decreases collagen deposition, and inhibits fibrogenic factor levels, thereby preventing the worsening of CCl_4_-induced HF [54]. From the mechanism perspective, CircCREBBP is the sponge for hsa-miR-1291, which up-regulates LEFTY2, thus alleviating liver fibrosis within TGF-β1-treated LX-2 cells, primary HSCs, and liver tissues from CCl_4_-induced HF mice, and liver fibrosis patients via the hsa-miR-1291/LEFTY2 axis [54]. 

##### CircFBXW4

CircFBXW4 (Mm9_circ_000338) is located on chromosome 19: 45705354-45715011 (510 nt) and consists of four exons (exon 2–5) at FBXW4 (the F box and WD 40 domain-containing protein 4) gene locus [21]. Chen et al. analyzed CircRNAs levels within HSCs in chronic liver fibrosis mice by CircRNAs-seq and reported that CircFBXW4 expression was significantly down-regulated during liver fibrogenesis, and it facilitated the HF recovery process [21]. The lower levels of CircFBXW4 are also present within peripheral blood from HF cases, in contrast to normal subjects. The expression of CircFBXW7 is consistently down-regulated within acute liver injury, CCl_4_, and BDL-induced HF mice. Further, the in vivo overexpression of CircFBXW4 by liver-specific administration of pHBAAV-circFBXW4 decreases collagen deposition and myofibroblast transdifferentiation, mitigates injuries to mouse liver fibrogenesis, and avoids inflammatory response, which reveals that CircFBXW4 resists against fibrosis in HF [21]. Moreover, the in vitro CircFBXW4 overexpression suppresses LX-2 cell activation, inhibits cell growth as well as DNA synthesis, causes cell cycle arrest, and ultimately induces LX-2 cell apoptosis. Eventually, as revealed from the luciferase reporter, fluorescence in situ hybridization and RNA pull-down assays conducted by Chen et al., CircFBXW4 bound to miR-18b-3p to modulate FBXW7 level as the miRNA sponge [21]. In view of its crucial roles in liver fibrosis and HSC activation, CircFBXW4 may be the candidate marker used to diagnose and treat the disease. 

##### CircPSD3

It is found that CircPSD3, which is derived from exons 4–8 of pleckstrin and the Sec7 domain that contains a three (PSD3) gene locus, markedly decreases within primary HSCs together with the liver tissues in CCl_4_-mediated HF mice. The in vivo CircPSD3 overexpression induced by AAV8-circPSD3 injection inhibits HSCs activation and proliferation and alleviates CCl_4_-induced HF, which can be evidenced by the decreased serum aspartate aminotransferase (AST) and alanine aminotransferase (ALT) levels, collagen deposition, liver hydroxyproline content, along with pro-inflammatory cytokine and pro-fibrogenic gene levels [12]. The in vitro CircPSD3 overexpression significantly decreased a-SMA and Col1α1 expression at mRNA and protein levels, while inhibiting HSC growth and activation. CircPSD3 is the mechanical sponge for miR-92b-3p, which can later promote Smad7 expression. Specifically, CircPSD3 alleviates the formation of liver fibrosis via miR-92b-3p/Smad7 axis, which suggests that CircPSD3 may be a potential biomarker for HF [12]. 

##### Hsa_circ_0070963

Hsa_circ_0070963 shows decreased expression within mice with CCl_4_-induced liver fibrosis, while the restoration of hsa_circ_0070963 expression can stop HSC activation and decrease type I collagen and α-SMA expression, both in vivo and in vitro [58]. In addition, as revealed by a luciferase reporter assay and rescue experimenthsa_circ_0070963 is confirmed to sponge miR-223-3p to inhibit HSC activation during liver fibrosis through regulating miR-223-3p and LEMD3 [58], while LEMD3 (called MAN1 as well), as the inner nuclear membrane protein, can inhibit the TGF-β receptor-mediated Smads pathway [58,80]. 

##### Hsa_circ_0004018

Li et al. initially studied the expression profile of CircRNA using a microarray, according to their results, hsa_circ_0004018 was significantly down-regulated within the fibrotic mouse liver. In particular, hsa_circ_0004018 down-regulation within HSCs was related to slow fibrosis progression. 

Subsequently, an in vitro study found that the hsa_circ_0004018 overexpression suppressed HSCs growth by blocking the cell cycle, as indicated by a higher G0/G1 phase ratio, and the decreased mRNA and protein expression of α-SMA and Col1α1 within HSCs, suggesting that hsa_circ_0004018 may inhibit HSC activation [22]. Further, according to an in vivo study, α-SMA and Col1α1 expression obviously decreases, and liver fibrosis is obviously alleviated following the administration of hsa_circ_0004018-expression lentivirus in mice with CCl_4_-mediated liver fibrosis [22]. Based on the above findings, telomerase-associated protein 1 (TEP1) is hsa-miR-660-3p’s downstream target. Thus, hsa_circ_0004018 is the potential hsa-miR-660-3p sponge, which can target and suppress TEP1 expression. TEP1 is one of the telomerase components, which assists in telomere maintenance [81], and previous studies have shown that telomeres in hepatocytes or immune cells may be involved in liver fibrosis [82]. In short, the hsa_circ_0004018/hsa-miR-660-3p/TEP1 axis is related to HSC activation and proliferation, which is the potential novel therapeutic strategy for liver fibrosis [22]. 

##### Mmu_circ_34116

Zhou et al. used a CircRNA microarray to investigate the differentially expressed CircRNAs using the mouse model of CCl_4_-mediated liver fibrosis, and discovered 14 CircRNAs with up-regulation and 55 with down-regulation in liver fibrosis tissues [59]. After exploring the expression profiles of mouse fibroblasts JS1 activated by TGF-β1, three CircRNAs were obtained, which displayed consistent expression patterns in both liver fibrotic tissues and JS1 fibroblasts. Among them, mmu_circ_33594 and mmu_circ_35216 expression increased significantly, while mmu_circ_34116 expression considerably declined, indicating that they may be associated with HSC activation [59]. 

Subsequently, mmu_circ_34116 expression was knocked down after transient mmu_circ_34116 siRNA transfection. According to their results, α-SMA mRNA and protein expression remarkably elevated, indicating that mmu_circ_34116 inhibits HSC activation [59]. Based on further bioinformatics analyses, circRNA_34116 contained miRNA response element (MRE) in miR-22, which showed competitive binding to miR-22, thus exerting indirect regulation on BMP7 target gene transcription [59]. Earlier studies have revealed that treating CCl_4_-mediated liver fibrosis mice with antisense miR-22 increases BMP7 expression, and remarkably suppresses the development of fibrosis. Therefore, miR-22 possibly enhances liver fibrosis development by suppressing BMP7 [83]. In brief, mmu_circ_34116 suppresses HSCs activation and fibrosis via mmu_circ_34116/miR-22/BMP7 axis [59]. 

##### CircDIDO1

CircDIDO1 (ID: hsa_circ_0061137) originates from two to six exons in the DIDO1 gene and is located at the chr20: 61537238-61545758 strand with a 1781-bp genomic length. As a stable circular transcript, CircDIDO1 is initially found to be primarily distributed in the cytoplasm of the human LX-2 cell line. However, Ma et al. observed the low and stable expression of CircDIDO1 within serum exosomes from liver failure cases and predicted that serum exosomal CircDIDO1 showed useful significance in diagnosing liver failure [60]. Thereafter, according to the in vitro research results, CircDIDO1 suppressed DNA synthesis and cell viability, reduced expression of a-SMA and collagen I pro-fibrotic markers, promoted cell apoptosis, and blocked the cell cycle of an LX-2 cell line by blocking the PTEN/AKT pathway [60]. Furthermore, when extracellular CircDIDO1 was added in exosomes obtained from mesenchymal stem cells (MSCs) and later transmitted into HSCs, the results indicated that extracellular CircDIDO1 restrained HSC activation by up-regulating PTEN, and it also inhibited the AKT pathway through sponging miR-141-3p. Mechanically, MSC-originated exosomal CircDIDO1 suppressed HSC activation and liver fibrosis development via miR-141-3p/PTEN/AKT pathway within human liver fibrosis [60], thus shedding new light on developing exogenous CircRNAs to prevent liver fibrosis [60]. 

##### CircMTO1

CircMTO1 originates from MTO1 (mitochondrial tRNA translation optimization 1 gene, hsa_circ_0007874) [61], which is recognized as the tumor suppressor of HCC [84]. CircMTO1, which can be detected on chr6:74175931-74176329 (circBase database), has decreased expression within liver tissues from cirrhosis cases, activated HSCs and the CCl_4_ initiated liver fibrosis model [61]. As suggested by Wang et al., CircMTO1 overexpression inhibited the TGF-β1-mediated HSC activation and cell cycle progression, suppressed α-SMA and type I collagen levels, and relieved liver fibrosis through targeting miR-17-5p and Smad7 [6]. In contrast, based on Zheng et al., miR-181b-5 activated HSC via the PTEN/AKT cascade [85], while Jin et al. further revealed that miR-181b-5p and CircMTO1 were co-distributed within the cytoplasm and interacted with each other. Thereafter, CircMTO1 suppressed HSC activation, which was almost completely restrained via PTEN or miR-181b-5p [61]. Consequently, CircMTO1 inhibited HSC activation through the miR-17-5p/Smad7 axis or miR-181b-5p-mediated PTEN expression, ultimately slowing down liver fibrosis. Such results indicate that CircMTO1 is the potential novel therapeutic target of liver fibrosis. Further, serum CircMTO1 expression is found to evidently decrease among patients with chronic hepatitis B (CHB), which shows negative relation to various stages of fibrosis. Similarly, CircMTO1 down-regulation predicts a dismal prognostic outcome, which is the candidate diagnostic biomarker for CHB patients [6]. 

##### CircRNA-0046367

Hepatic steatosis is characterized by lipid peroxidation and the accumulation of triglyceride (TG), thus resulting in non-alcoholic steatohepatitis, cirrhosis, liver fibrosis, or HCC. As discovered by Guo et al., the cytoplasmic lipid droplets and intracellular TG levels increased significantly in free fatty acid (FFA)-mediated steatosis HepG2 cells, while the expression of CircRNA-0046367 was significantly down-regulated [63]. Next, they revealed through bioinformatic analysis that only miR-34a was related to both CircRNA-0046367-targeted miRNA and steatosis, and a dual-luciferase reporter assay further confirmed that CircRNA-0046367 bound to miR-34a in a complementary manner [63]. The administration of CircRNA-carrying vectors into steatosis HepG2 cells induced the expression of CircRNA-0046367, thereby having an antagonizing function in miR-34a. Moreover, recovery of CircRNA-0046367 in steatosis HepG2 cells resulted in the significantly up-regulated translation and transcription levels of peroxisome proliferator-activated receptor α (PPARα), which promoted transcription of carnitine palmitoyltransferase 2 (CPT2) as well as an acyl-CoA binding domain containing three (ACBD3). Overall, CircRNA-0046367 prevented the hepatotoxicity of lipid peroxidation associated with steatosis, which manifested as mitochondrial dysfunction, apoptosis, and growth arrest [63]. All these results suggest that there is a CircRNA-0046367/miR-34a/PPARα regulatory system to facilitate the progression of hepatic steatosis. 

##### Mmu_circ_0000623

Numerous studies have suggested that exosomes derived from adipose mesenchymal stem cells (ADSCs) may be adopted for delivering CircRNA to treat liver fibrosis [86]. As demonstrated by Zhu et al., mmu_circ_0000623 level decreased within CCl_4_-treated mouse livers using high-throughput circRNA microarrays. Furthermore, in vivo and in vitro experimental results suggest that ADSCs-derived exosomes, particularly mmu_circ_0000623-modified ADSCs-derived exosomes, remarkably inhibited CCl_4_-mediated liver fibrosis through the activation of autophagy by interacting with miR-125/ATG4D. Both proteins related to autophagy as well as autophagic plaques with positive ATG4D expression were under regulation via mmu_circ_0000623/miR-125 [62]. 

##### CircRNA-608

CircRNA sequencing is performed on the isolated mouse PHSCs transfected with si-PINK1, after which, 37 CircRNAs are validated with differential expression in the si-PINK1 group. Among them, CircRNA-608 is significantly down-regulated in NASH-related liver fibrosis mice and siPINK1-LX-2 cells. Based on the TargetScan database (http://www.targetscan.org, accessed on 1 April 2023), only miR-222 is predicted to possess binding sites of PINK1. The overexpression of CircRNA-608 leads to the inhibited expression of miR-222, while sh-circRNA-608 induces miR-222 up-regulation within LX-2 cells. Thus, miR-222 is proved to be negatively regulated by CircRNA-608. Subsequently, miR-222 mimics (or inhibitors) are introduced into LX-2 cells treated with PA (palmitic), and it is shown that miR-222 has a negative regulatory effect on the expression of PINK1 in HSCs. Moreover, PINK1 mRNA level markedly decreases following sh-circRNA-608 transfection in PA-treated LX-2 cells, while further miR-222 mimic treatments aggravate this effect. According to the above findings, CircRNA-608 enhances PINK1-regulated mitophagy to slow down NASH-associated liver fibrosis by inhibiting miR-222 within lipotoxic HSCs, which provides novel insights into the pathogenic mechanism of NASH-associated liver fibrosis [64]. 

##### LNCPINT-Derived CircRNAs

The NAFLD model is constructed in mice fed with a high-fat and high-cholesterol (HFHC) diet, after which it is found that 28 CircRNAs show remarkable down-regulation within NAFLD mouse livers. Of those downregulated CircRNAs, long intergenic non-protein coding RNA, and P53-induced transcript (LNCPINT)-derived CircRNAs (CircRNA-0001452, CircRNA-0001453, and CircRNA-0001454) target miR-466i-3p and miR-669c-3p simultaneously. The loss of these CircRNAs promotes miR-466i-3p and miR-669c-3p activation, thereby inactivating the AMPK pathway by suppressing AMPK-α1 expression. Inhibition of the AMPK pathway promotes the transcription and translation of lipogenic genes, including fatty acid synthase (FASN) and encoding sterol regulatory element-binding protein 1 (SREBP1) within hepatocytes, finally promoting hepatic steatosis. Furthermore, LNCPINT-derived CircRNAs levels are negatively related to TG contents in the liver. Collectively, LNCPINT-derived CircRNAs loss may be related to NAFLD through inactivating the AMPK pathway in a miR-466i-3p- and miR-669c-3p-dependent manner [65]. 

##### MecciRNAs

After downloading and analyzing the GSE134146 microarray dataset, Liu et al. confirmed that there were 20 CircRNAs with up-regulation and eight with down-regulation in the HSCs of NASH patients. Among them, mecciRNAs (mitochondrial genome) occupied 50% of CircRNAs with down-regulation, which were primarily located in mitochondria. Based on CircMINE and GSE46300 databases, there were two mecciRNAs (hsa_circ_0089761, hsa_circ_0089763) predicted to be ceRNAs for regulating fibrosis-related signals. The in vitro contents decreased in LPS-activated LX-2 in contrast to wild-type LX-2, while the expression of miR-642a-5p, miR-670-3p, miR-1248, and miR-1224-3p increased, similarly, Smad2, Smad3, and c-Myc protein levels remarkably increased within the activated HSCs [66]. However, another two mecciRNAs (hsa_circ_0089762, hsa_circ_0008882) had small levels, thus, they might be the molecular scaffolds for regulating specific complex activities or be molecular chaperones in mitochondria-imported protein folding [67,68]. 

### 4.2. CircRNAs Are Involved in Promoting Liver Fibrosis

Certain CircRNAs show up-regulation within HSCs and liver fibrotic tissues and promote liver fibrosis development via several mechanisms (e.g., playing a role of miRNA sponges, combining with functional miRNAs, and modulating transcription and post-transcription levels of genes) (Table 1) [4,55,70]. The above CircRNAs with up-regulation are the potential diagnostic markers and therapeutic targets of liver fibrosis. 

#### 4.2.1. CircRNAs Are Related to Liver Fibrosis by Playing a Role of miRNAs-Binding CeRNAs

##### CircUbe2k

CircUbe2k (mmu_circ_0001350) is located on chromosome chr5: 65957227-65985787 (465nt), and consists of five exons (exon 2–6) at Ube2k gene locus [55]. CircUbe2k expression is shown to be significantly enhanced in TGF-β1-stimulated LX-2 cells and CCl_4_-mediated liver fibrosis [55]. However, when CircUbe2k expression decreases by siRNA-circUbe2k transfection within the activated LX-2 cells, α-SMA and Col1α1 mRNA levels subsequently decrease, and LX-2 cell viability and proliferation are strongly suppressed. Further, injection with AAV-siRNA-circUbe2k leads to the inhibition of α-SMA, Col1α1, TGF-β1, and TIMP-1 expression. Specifically, collagen and ECM deposition are reduced in mice with CCl_4_-mediated liver fibrosis, which thereby suppresses liver fibrosis [55]. From the mechanism perspective, CircUbe2k, as the miRNA sponge, has the capacity to bind to miR-149-5p, so as to regulate the TGF-β2 level. This indicates that it can promote HSCs activation and HF progression via CircUbe2k/miR-149-5p/TGF-β2 axis [55]. 

##### CircPWWP2A

The expression of CircPWWP2A (corresponding to hsa_circ_0074837 in humans and mmu_circ_0000254 in mice) significantly increases in LPS- and TGF-β-stimulated HSCs, as well as in liver tissues in mice with liver fibrosis. Further mechanistic studies demonstrate that CircPWWP2A promotes HSC growth and activation as a sponge of miR-203 and miR-223, which then promotes Fstl1 and TLR4 levels, respectively, and eventually promotes liver fibrosis [4]. 

##### CircRNA-4099

As suggested by a number of articles, liver fibrosis, usually resulting from oxidative stress [5], represents the eventual common pathway underlying hepatitis of different causes. Hepatitis accounts for one liver dysfunction type that typically indicates various levels of damage to liver cells induced by a variety of pathogenic factors [87]. 

As an oxidant, H_2_O_2_ can significantly increase the cellular ROS levels and α-SMA, type I, and type III collagen production within L02 human hepatocytes, thus inhibiting L02 cell viability dose-dependently and inducing L02 cell fibrosis and injury [69]. The L02 cells are treated with different H_2_O_2_ dilutions for mimicking hepatitis in vitro, and the results demonstrate that H_2_O_2_ induces severe cell damage and CircRNA-4099 overexpression. Furthermore, the depletion of CircRNA-4099 alleviates the H_2_O_2_-induced cell damage, while its overexpression exhibits the opposite effects [69]. Further mechanistic research implies that CircRNA-4099 aggravates H_2_O_2_-mediated damage by suppressing miR-706 by activating p38MAPK and keap1/Nrf2 within L02 cells. This suggests that CircRNA-4099 may be a key therapeutic target for hepatitis and fibrosis [69]. 

##### CircRNA-0067835

Thymosin β4 (Tβ4) is the 43-amino acid polypeptide with a high conservation degree and the most abundant actin-sequestering protein in cells, which promotes cell growth and survival via the PI3K/AKT signaling pathway [85,88,89]. Zhu et al. applied a circRNA microarray in investigating Tβ4-associated CircRNAs and identified altogether 644 CircRNAs with differential expression in Tβ4-deficient LX-2 cells compared with control cells [70]. Further studies revealed that the CircRNA-0067835 level was remarkably elevated within Tβ4-siRNA LX-2 cells relative to controls, and the CircRNA-0067835 knockout obviously reduced cell growth by inducing G1 arrest while enhancing apoptosis [70]. Moreover, the CircRNA-0067835 level was markedly elevated within the CCl_4_-treated mouse livers. Finally, CircRNA-0067835 was demonstrated to promote liver fibrosis development by sponging miR-155 for promoting FOXO3a and AKT expression [70]. 

##### CircRSF1 and Hsa_circ_0071410

Radioactive liver disease (RILD) has been recognized to be the main complication secondary to radiotherapy for liver cancer [90], which may induce liver failure or even death in some severe cases [91]. The abnormal expression of CircRNA is possibly related to cellular responses to radiation and the biological process of liver fibrosis. Chen et al. elucidated that the Ras-related C3 botulinum toxin substrate 1 (RAC1) and miR-146a-5p levels markedly increased within the human HSCs (LX-2) irradiated with 8 Gy X-rays. RAC1 overexpression remarkably promoted proinflammatory factor production, and α-SMA and collagen I levels, by promoting NF-κB p65 phosphorylation and Bcl-2 expression within irradiated LX-2 cells. However, the overexpression of miR-146a-5p induced via miR-146a-5p mimic transfection repressed RAC1 and Bcl2 levels and decreased the synthesis of α-SMA and type I collagen through inhibiting the phosphorylation of NF-κB p65, JNK1, and Smad1. Therefore, miR-146a-5p reversed the regulation of RAC1 in HSCs [71]. Based on bioinformatics analysis, cricRSF1 (hsa_circ_0023706) contained three MREs of miR-146a-5p [92]. Since CircRSF1 was up-regulated in irradiated LX-2 cells, Chen et al. further co-transfected the CircRSF1 expression vector and miR-146a-5p into LX-2 cells. As a result, CircRSF1 reduced inhibition of miR-146a-5p on the RAC1-triggered activation of NF-κB and JNK/Smad2 pathways, thereby promoting pro-inflammatory factor expression and production, increasing fibrosis marker expression, and finally stimulating the activation of HSCs [71]. Collectively, these results indicate that there is a function regulatory axis involving CircRSF1, miR-146a-5p, and RAC1 within the irradiated HSCs. 

Furthermore, as clarified by Chen et al., miR-9-5p expression evidently decreased within the irradiated LX-2 cells, while hsa_circ_0071410 silencing up-regulated miR-9-5p [57]. Meanwhile, hsa_circ_0071410 knockdown significantly reduced α-SMA protein and mRNA expression within the irradiated LX-2, and inhibited the irradiated LX-2 cell growth, probably through inhibiting miR-9-5p. These results indicate that hsa_circ_0071410 promotes irradiated fibrosis via miR-9-5p. 

##### CircTUBD1

A three-dimensional (3D) LX-2 cell culture model is developed in vitro, and CircRNAs with differential expression within the LX-2 cells after 8 Gy X-ray radiation are screened using a CircRNA microarray. It is elucidated that CircTUBD1 (hsa_circ_0044897) displays significant up-regulation, and alterations in its expression conform to LX-2 cell activation. The in vitro results further indicate that CircTUBD1 knockdown significantly suppresses LX-2 cell activation and apoptosis induced by radiation. The in vivo experiments reveal that the CircTUBD1 knockout partially reduces fibrosis marker levels and relieves the early liver fibrosis resulting from radiation within a radiation-induced liver fibrosis (RILF) mouse model. Subsequently, multi-function experiments verify that CircTUBD1 regulates LX-2 cell activation and responses to fibrosis resulting from irradiation by the CircTUBD1/micro-203a-3p/Smad3 axis. Since the down-regulation of CircTUBD1 genes alleviates early radiation-induced liver fibrosis in mice, it may offer a new treatment to mitigate RILF progression [72]. 

##### Hsa_circ_0072765

Hsa_circ_0072765 is located on chr5: 68471223-68471364 and consists of the CCNB1 gene. As indicated by Jin et al., the hsa_circ_0072765 level significantly elevated within TGF-β1-exposed LX-2 cells and exosomes. Next, they synthesized a si-hsa_circ_0072765 vector and transfected it into LX-2 cells to knock down the hsa_circ_0072765 expression. According to their results, hsa_circ_0072765 knockdown repressed the growth, invasion, and activation of TGF-β1-treated LX-2 cells. Through searching the circinteractome, miR-197-3p possessed the complementary sequence of hsa_circ_0072765, while based on further verification experiments, hsa_circ_0072765 regulated the growth, invasion, and activation of TGF-β1-treated HSCs by targeting miR-197-3p. Through Targetscan analysis, TRPV3 was identified as miR-197-3p’s target gene, and hsa_circ_0072765 overexpression within LX-2 cells increased TRPV3 protein expression, whereas the hsa_circ_0072765 gene knockout reversed this effect. In brief, hsa_circ_0072765 promotes TGF-β-mediated liver fibrosis development by reducing miR-197-3p level while inducing the TRPV3 level [73]. 

##### CircMcph1

CircMcph1 originates from the mouse chromosome 8:18668945-18689059 (279 nt), and CircMcph1 is remarkably up-regulated within primary Kupffer cells and liver tissues in HF mice. The in vivo administration of AAV8-GFP-circMcph1 leads to the decreased expression of tumor necrosis factor-α (TNF-α), IL-1β, and monocyte chemoattractant protein-1(MCP-1). Further, hepatic fat vacuoles, collagen deposition, as well as inflammatory cell infiltration are relieved. LPS and interferon-γ (IFN-γ) are employed to establish an in vitro inflammatory injury model in mouse RAW264.7 cells, as a result, CircMcph1 expression increases, while TNF-α, IL-1β, and MCP-1 mRNA expression decrease and cell apoptosis are reduced. However, CircMcph1 knockout can reverse these effects. Subsequently, the following predictive analysis and functional experiments prove that CircMcph1 regulates Irak2 expression by acting as a sponge of miR-370-3p. Collectively, the above findings reveal that Irak2 overexpression reverses the anti-inflammation activity induced by CircMcph1 knockout, probably through targeting the miR-370-3p/Irak2 axis [74]. 

##### CircRNA-0008494

After identifying 363 CircRNAs with up-regulation and 635 with downregulation within human HF tissues via RNA sequencing (RNA-seq), Li et al. revealed (using the miRanda algorithm) that only CircRNA-miRNA-mRNA network of CircRNA-0008494 contained several HF-associated genes (especially distinct collagens). Consistently, CircRNA-0008494 and the fibrosis markers α-SMA and Col1α1 significantly increased within TGF-β1-treated LX-2 cells. Thereafter, the lentiviral constructs LV-circRNA-0008494 were applied to infect LX-2 cells, as a result, cell migration capacity and α-SMA and Col1α1 expression dramatically decreased, while apoptosis rate significantly increased. However, the suppressed expression of CircRNA-0008494 restrained the growth, invasion, and activation, and enhanced HSCs apoptosis. Subsequently, the miRanda algorithm, dual-luciferase reporter assay, and multi-functional experiments confirmed that CircRNA-0008494 modulated HSCs activation via CircRNA-0008494/miR-185-3p/Col1α1 axis [75]. 

## 5. Potential Value of CircRNAs in the Treatment of Liver Fibrosis

Many microarrays or high-throughput sequencing results have suggested that many CircRNAs are differentially expressed in liver fibrotic tissues. However, most of them act as miRNA sponges, and a small number function as RNA scaffolds (Table 1). Several CircRNAs are down-regulated in liver fibrotic tissues and HSCs, which have been demonstrated to inhibit liver fibrosis. However, some CircRNAs are found to be up-regulated, which may promote the progression of liver fibrosis via multiple mechanisms (Table 1). Liver fibrosis is the precursor of liver cirrhosis and HCC. Therefore, timely diagnosis and treatment in the stage of liver fibrosis will prevent the disease from further developing into liver cirrhosis and even HCC. Fortunately, it is well known that CircRNAs are stable, and their levels in liver tissues can be easily quantified by quantitative RT-PCR [54]. Furthermore, some CircRNAs have been demonstrated to be more accurate in predicting HCC than alpha-fetoprotein [93], which will bring new hope for the application of CircRNA biomarkers in the diagnosis of liver fibrosis. Meanwhile, Hu et al. found that CircCHD2 may have the potential to be a therapeutic target for liver cirrhosis [94]. In addition, several strategies for the modulation of CircRNA expression in HCC, have been successfully implemented in preclinical models [95,96]. Therefore, it is possible to detect CircRNAs expression levels in liver fibrosis, and to consider CircRNAs as the diagnostic biomarkers for liver fibrosis, and the modulation of CircRNA expression may be the new therapeutic strategy for liver fibrosis in the future. Moreover, since CircRNAs have been detected in vesicles [29], they can be transported throughout the body to function as signaling molecules. Therefore, CircRNAs may be used as ideal vehicles for drug delivery and release. Considering that CircRNAs themselves may also be used for the treatment of fibrosis, the combination of CircRNAs and their delivered drugs will be a very promising therapeutic strategy for the treatment of fibrosis. However, most of the differentially expressed CircRNAs in the above studies are found in liver tissue, and their expression changes and stability in plasma or serum remain to be verified. Furthermore, the diagnostic value of CircRNAs for liver diseases still requires further exploration because of the heterogeneity of liver disease. Meanwhile, the immunogenicity of CircRNAs may be a major obstacle in the application of CircRNA-directed therapies. Therefore, there are still significant challenges with regard to the reliability and safety of these diagnostic and therapeutic strategies, which will require further study.

## 6. Conclusions and Future Perspectives

CircRNA is a critical regulatory factor in liver fibrosis, where aberrant CircRNA levels are associated with HSC activation and proliferation. Thus, this review illustrates the roles and mechanisms of CircRNAs, and summarizes some of the latest studies of primary CircRNAs involved in regulating liver fibrosis. Two major models of action are related to the roles of CircRNAs in modulating liver fibrosis, namely, by serving as microRNA sponges or RNA scaffolds. Although protein sponge, protein scaffold, and translational value of CircRNAs seem to be significant in HCC and Duchenne muscular dystrophy [45,97], whether these modes of action exist in liver fibrosis remains unclear. Therefore, we fully expect that more regulatory functions and action modes of this large class of RNA molecules will emerge in the process of liver fibrosis, which will bring a new revolution in liver fibrosis-related research. In addition, most of the current studies concerning the mechanisms of CircRNAs in liver fibrosis remain at the cellular and animal levels, and the transformation into clinical applications will be confronted with significant challenges and will become a hot topic in future research. In a word, this review will help to understand the functions of various CircRNAs and the associated mechanisms underlying liver fibrosis development and assist in applying CircRNA-directed diagnoses and therapies in liver fibrosis in the future.

## Figures and Tables

**Figure 1 biomolecules-13-00940-f001:**
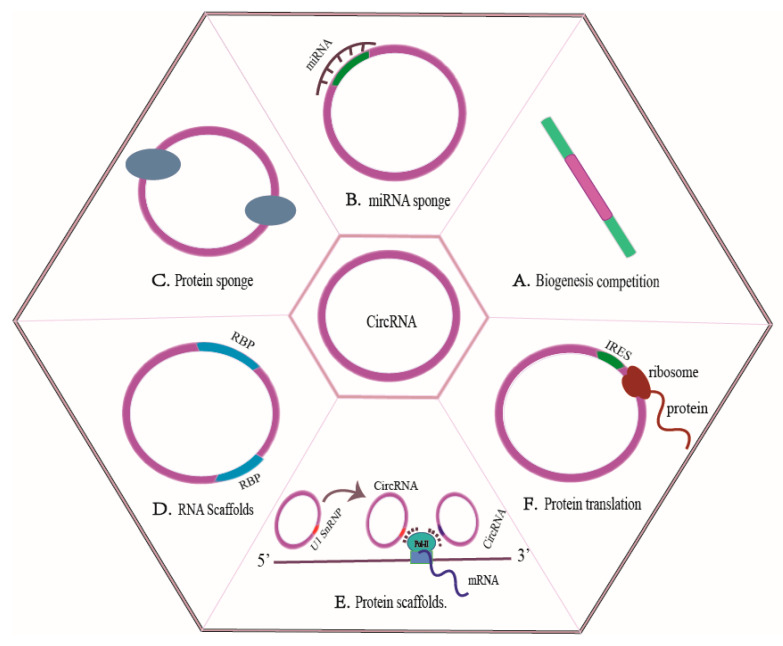
CircRNAs mechanisms of action. (**A**) The interplay between backsplicing and canonical splicing for the same splice site selection that can ultimately affect pre-mRNA splicing. (**B**) CircRNAs as ceRNAs that modulate the bioavailability of miRNAs. (**C**) CircRNAs can function as decoys or sponges for proteins that bind to proteins. (**D**) CircRNAs as RNA scaffolds. (**E**) Circular RNAs as protein scaffolds. (**F**) CircRNAs regulate protein translation.

**Figure 2 biomolecules-13-00940-f002:**
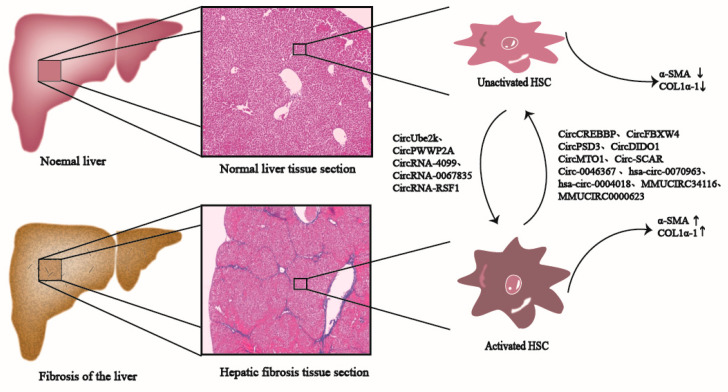
CircRNAs participate in the liver fibrosis process. CircRNAs, such as CircUbe2k, CircPWWP2A, and CircRNA-4099 promote the activation of HSCs in normal liver tissues, generate pro-fibrosis factors such as α-SMA and Col1α1, and ultimately promote the development of liver fibrosis; CircRNAs, such as CircCREBBP, CircFBXW4, and CircPSD3 inhibit the activation of HSCs, reduce the production of pro-fibrosis factors such as α-SMA and Col1α1, and eventually inhibit liver fibrosis.

**Table 1 biomolecules-13-00940-t001:** CircRNAs are involved in regulating liver fibrosis.

CircRNAs	Targets	Biological Function	Expression Pattern	Ref.
Inhibition of liver fibrosis				
CircRNA SCAR	ATP5B	CircRNA SCAR overexpression clearly inhibit the contractility, collagen and α-SMA expression in NASH fibroblasts, thereby alleviating the fibrotic phenotype.	decrease	[46]
CircCREBBP	hsa-miR-1291	CircCREBBP, as a sponge for hsa-miR-1291, promotes the expression of LEFTY2 and inhibits the activation of hepatic stellate cells.	decrease	[54]
CircFBXW4	miR-18b-3p	CircFBXW4, as a miRNA sponge for miR-18b-3p, to regulate the expression of FBXW7, inhibiting the activation of hepatic stellate cells.	decrease	[21]
CircPSD3	miR-92b-3p	CircPSD3 can inhibit HSC activation and proliferation by targeting the miR-92b-3p/Smad7 axis to alleviate the formation of liver fibrosis.	decrease	[12]
Hsa_circ_0070963	miR-223-3p	Hsa_circ_0070963, as a miR-223-3p sponge, inhibites HSC activation via regulation of miR-223-3p and LEMD3 in liver fibrosis.	decrease	[58]
Hsa_circ_0004018	hsa-miR-660-3p	Hsa_circ_0004018 acts as a sponge of hsa-miR-660-3p, and then targetedly suppress the expression of TEP1 to inhibit the activation of HSCs.	decrease	[22]
Mmu_circ_34116	miR-22	Mmu_circ_34116 inhibits the activation of HSCs and fibrosis by mmu_circ_34116/miR-22/BMP7 signal axis.	decrease	[59]
CircDIDO1	miR-141-3p	Extracellular circDIDO1 restraines HSC activation by miR-141-3p/PTEN/AKT pathway in liver fibrosis.	decrease	[60]
CircMTO1(Hsa_circ_0007874)	miR-17-5p, miR-181b-5p	CircMTO1 can inhibit TGF-β1-induced HSC activation by targeting miR-17-5p and Smad7 or miR-181b-5p/PTEN/AKT cascade.	decrease	[6,61]
CircRNA-0046367	miR-34a	CircRNA-0046367 promotes the expression of CPT2 and ACBD3 by promoting PPARα through CircRNA-0046367/miR-34a/PPARα axis, and reduces steatosis of hepatocytes.	decrease	[63]
Mmu_circ_0000623	miR-125	Mmu_circ_0000623-modified ADSCs, significantly suppressed CCl4-induced liver fibrosis by promoting autophagy activation through interacting with the miR-125/ATG4D.	decrease	[62]
CircRNA-608	miR-222	CircRNA-608 might promote PINK1-mediated mitophagy to slow down NASH-related liver fibrosis though inhibiting miR-222 in lipotoxic HSCs.	decrease	[64]
LNCPINT-derived CircRNAs (Circ_0001452, Circ_0001453, and Circ_0001454)	miR-466i-3p, miR-669c-3p	Loss of LNCPINT-derived CircRNAs may underlie NAFLD via miR-466i-3p and miR-669c-3p-dependent inactivation of the AMPK signaling pathway.	decrease	[65]
MecciRNAs (Hsa_circ_0089761 and Hsa_circ_0089763)	miR-642a-5p, miR-1248, miR-670-3p and miR-1224-3p	Hsa_ circ_0089761 and hsa_circ_0089763 could function as competing for endogenous RNAs (ceRNAs) to regulate fibrosis-related signals.	decrease	[66]
MecciRNAs (Hsa_circ_0089762 and Hsa_circ_0008882_		Hsa_circ_0089762 and hsa_circ_0008882 might act as molecular scaffolds to regulate specific complex functions or serve as molecular chaperones in the folding of mitochondria-imported proteins.	decrease	[67,68]
Promotion of liver fibrosis				
CircUbe2k(Mmu_circ_0001350)	miR-149-5p	CircUbe2k could promote the activation of HSCs and the progress of HF through the CircUbe2k/miR-149-5p/TGF-β2 axis.	increase	[55]
CircPWWP2A	miR-203, miR-223	CircPWWP2A promoted the activation and proliferation of HSCs via sponging miR-203 and miR-223, subsequently enhanced the expression of Fstl1 and Tlr4, respectively, and eventually promoted hepatic fibrosis.	increase	[4]
CircRNA-4099	miR-706	CircRNA-4099 aggravated H_2_O_2_-induced damage and fibrosis by inhibiting miR-706 through triggering keap1/Nrf2 and p38MAPK in L02 cell lines.	increase	[69]
CircRNA-0067835	miR-155	CircRNA-0067835 can promote liver fibrosis progression by acting asa sponge of miR-155 to promote FOXO3a and AKT expression in LX-2 cells.	increase	[70]
CircRSF1	miR-146a-5p	CircRSF1 can promote the activation of HSC cells by the CircRSF1/miR-146a-5p/RAC1 axis in irradiated LX-2 cells.	increase	[71]
Hsa_circ_0071410	miR-9-5p	Hsa_circ_0071410 inhibits the proliferation of LX-2 by Hsa_circ_0071410/miR-9-5p in irradiated LX-2cells.	increase	[57]
CircTUBD1(Hsa_circ_0044897)	miR-203a-3p	CircTUBD1 regulated the activation and fibrosis response of LX-2 cells induced by irradiation via circTUBD1/miR-203a-3p/Smad3 axis.	increase	[72]
Hsa_circ_0072765	miR-197-3p	Hsa_circ_0072765 promotes the progression of HSCs and liver fibrosis induced by TGF-β through decreasing the expression of miR-197-3p and inducing the expression of TRPV3.	increase	[73]
CircMcph1	miR-370-3p	CircMcph1 could regulate the expression of Irak2 by sponging miR-370-3p.	increase	[74]
CircRNA-0008494	miR-185-3p	CircRNA-0008494 can regulate activation, proliferation, migration and apoptosis of HSCs by CircRNA-0008494/miR-185-3p/Col1a1 axis.	increase	[75]

Several CircRNAs (e.g., CircCREBBP, CircFBXW4, CircPSD3, hsa_circ_0070963, hsa_circ_0004018, mmu_circ_34116, CircDIDO1, CircMTO1 and mmu_circ_0000623 et al.) have been demonstrated to inhibit liver fibrosis. Some CircRNAs are upregulated in liver fibrotic.tissues and HSCs, which promote the progression of liver fibrosis through multiple mechanisms (e.g., acting as miRNA sponges, binding to functional miRNAs, and regulating gene expression at the transcriptionalor post-transcriptional level).

## Data Availability

Not applicable.

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
