# Peer review of "Overview of CircRNAs Roles and Mechanisms in Liver Fibrosis"

_biomolecules, 2023, doi:10.3390/biom13060940_

Round 1

Reviewer 1 Report

In this review the authors described circRNAs founded and participated in the development of liver fibrosis. Several papers were published with the similar topic but circRNAs were analysed in different liver diseases not only the fibrosis. The main advantage of the paper is a deep focus on the fibrosis and the full list of circRNAs founded specific for the fibrosis. But this review just summarize the published data without any additional analysis and prognosis for the diagnosis and treatment as it was mentioned in the abstact. Please, add the section about the potential of circRNAs for therapy of the fibrosis and upgrade the conclusion section with the analysis of the listed functions. 

No comments

Author Response

Reviewer #1: In this review the authors described CircRNAs founded and participated in the development of liver fibrosis. Several papers were published with the similar topic but CircRNAs were analysed in different liver diseases not only the fibrosis. The main advantage of the paper is a deep focus on the fibrosis and the full list of CircRNAs founded specific for the fibrosis. But this review just summarizes the published data without any additional analysis and prognosis for the diagnosis and treatment as it was mentioned in the abstract. Please, add the section about the potential of CircRNAs for therapy of the fibrosis and upgrade the conclusion section with the analysis of the listed functions.

Answer: Thanks for your constructive suggestion. According to your advice, we add a section about the potential value of CircRNA in the treatment of liver fibrosis, which is presented in the fourth part of the manuscript. And we further upgrade our conclusions by further analyzing the functions of CircRNAs.

The added section is displayed as follows:

  1. Potential value of CircRNA in the treatment of liver fibrosis

Many microarrays or high-throughput sequencing results have suggested that many CircRNAs are differentially expressed in liver fibrotic tissues. Several CircRNAs (such as CircCREBBP, and CircFBXW4) are down-regulated in liver fibrotic tissues and HSCs, which have been demonstrated to inhibit liver fibrosis. However, some CircRNAs are found to be up-regulated, which promote the progression of liver fibrosis via multiple mechanisms (Table 1). Consequently, these validated differentially expressed CircRNAs in liver fibrosis may serve as the diagnostic biomarkers and therapeutic targets. Fortunately, it is well known that CircRNAs are stable, and their levels in serum and liver tissues can be easily quantified by quantitative RT-PCR. Thus far, some CircRNAs have been demonstrated to be more accurate in predicting HCC than alpha-fetoprotein. In addition, several strategies for the modulation of CircRNA expression in HCC, have been successfully implemented in preclinical models. Therefore, it is possible to detect CircRNAs expression levels in liver fibrosis, and to consider CircRNAs as the diagnostic biomarkers for liver fibrosis, and the modulation of CircRNA expression may be the new therapeutic strategy for liver fibrosis in the future.

The conclusions after upgrading are as follows:

CircRNAs are the critical regulatory factors in liver fibrosis, where aberrant CircRNA levels are associated with HSC activation and proliferation. Thus, this review illustrates the roles and mechanisms of CircRNAs, and summarizes some of the latest studies of primary CircRNAs involved in regulating liver fibrosis. This can help to understand functions of various CircRNAs and the associated mechanisms underlying liver fibrosis development. Two major models of action are related to the roles of CircRNAs in modulating liver fibrosis, namely, by serving as microRNA sponges or RNA scaffolds.

Although protein sponge, protein scaffold, and translational value of CircRNAs seem to be significant in HCC and Duchenne muscular dystrophy, whether these modes of action exist in liver fibrosis remains unclear. In addition, most of the current studies concerning the mecha-nisms of CircRNAs in liver fibrosis remain at the cellular and animal levels, and how to promote the transformation into clinical application will become a hot topic in future research. Moreover, CircRNAs are stable and can also be detected in body fluids such as serum, which will bring new hope for the diagnosis of liver fibrosis and assists in the development of CircRNAs as candidate therapeutic targets for a variety of liver diseases. Nonetheless, there are still significant challenges with regard to the safety and reliability of these therapeutic strategies, which will require further study. This study assists in applying CircRNA-directed therapies in suppressing liver fibrosis and liver diseases in the future.

Moderate editing of English language:

Answer: Thanks for your suggestions on the language of our manuscript. According to your advice, we invited Frank Boehm to check the tenses and grammar through the WHOLE manuscript, and believe that the language is now acceptable for the review and publication process.

Frank Boehm: a Canadian English speaker, who has been working on academic manuscript editing for Lakehead University, the University of Guelph, as well as other universities in Canada and China, for more than 20 years. He has also published one book (Nanomedical Device and Systems Design Challenges, Possibilities, Visions) (2013) on the topic of nanomedicine, with three upcoming books in progress.

Reviewer 2 Report

1The authors wrote a review article about circrnas in liver fibrosis. The text and the structure still need review and editing.

     1. The authors should also address RNA editing as a possible mechanism of circrna regulation in human diseases such as cancer or cardiac insufficiency/fibrosis. It has already been shown that RNA editing decreases in diseased tissues such as liver, heart or brain tissue and this leads to increasing circrna levels. Here the following citations could be inserted:

https://doi.org/10.1038/s41467-022-29138-2   10.1007/s00395-022-00940-9

2.       Unfortunately, most of the review consists of lists of individual circRNAs or summaries of their respective circrna function. What is missing here is an overview in which the circrnas of liver fibrosis are assigned to higher-level processes, such as a table or something similar.

3.       The abstract is not written very fluently. The functions should be better summarized and "etc." should not occur here. Also, a general overview should be given here and not a detailed description of individual circrnas.

4.       The introduction should also start with a general setup and not with secondary aspects of the disease.

TThe overall English is not sufficient, the review needs to be corrected by a native speaker. For example, it is not a “study” but a review.

Author Response

Reviewer #2:The authors wrote a review article about CircRNAs in liver fibrosis. The text and the structure still need review and editing.

  1. The authors should also address RNA editing as a possible mechanism of CircRNA regulation in human diseases such as cancer or cardiac insufficiency/fibrosis. It has already been shown that RNA editing decreases in diseased tissues such as liver, heart or brain tissue and this leads to increasing CircRNA levels. Here the following citations could be inserted:https://doi.org/10.1038/s41467-022-29138-2   10.1007/s00395-022-00940-9.

Answer: Thanks for your constructive suggestion. After reading the reference provided by you, we found that RNA editing from adenosine-to-inosine (A-to-I) is reduced in heart failure, and CircRNA formation is increased by regulating the RNA stability of ADAR2, which provides further insight in the regulation of CircRNA formation by RNA editing. In addition, we found that the splicing mode of pre-mRNA, RNA-binding proteins, intronic repeats sequences and the state of cell division are also factors affecting the formation or levels of circRNAs. According to your advice, we have added the above-mentioned contents and references in lines 79-96 of the manuscript, and the added contents are as follows:

On the other hand, the splicing mode of pre-mRNA can directly compete with CircRNA biogenesis, thus affecting the CircRNA levels. For example, the alternative splicing factor, quaking (QKI), can regulate circRNA biogenesis via binding sites in introns during epithelial-mesenchymal transition (EMT), and RNA-binding protein 20 (RBM20) is reported to modulate circRNA production from the Titin gene by excluding specific exons from the pre-mRNA. On the other hand, the production of SCD-circRNA 2 is dynamically regulated by RNA-binding protein 3 (RBM3) in HCC cell lines, but it remains unclear how RBM3 affects the alternative splicing of SCD-circRNA 2. Furthermore, intronic repeats sequences are also factors affecting the biogenesis of circRNAs in animals. It has also been discovered that adenosine deaminase acting on RNAs (ADARs) play a role of potent regulators of circular transcriptome in different types of cancer cells, including esophageal, breast, colon, liver, and gastric cancers. Moreover, RNA editing from adenosine-to-inosine (A-to-I) is reduced in heart failure, and CircRNA formation is up-regulated by regulating the RNA stability of ADAR2, which provides further insight in the regulation of CircRNA formation by RNA editing. Additionally, the state of cell division affects the CircRNA levels, such as the higher levels in slowly dividing cells whereas the lower levels in rapidly dividing cells. Therefore, CirRNA production is affected by the transcriptional level, post-transcriptional level and cell proliferation status.

  1. Unfortunately, most of the review consists of lists of individual CircRNAs or summaries of their respective CircRNA function. What is missing here is an overview in which the CircRNAs of liver fibrosis are assigned to higher-level processes, such as a table or something similar.

Answer: Thanks for your constructive suggestion. In fact, the overview of the CircRNAs of liver fibrosis had been summarized and displayed in the Table 1 of the last submitted manuscript. According to your advice, we added more table notes under the Table 1, and hope this time it would satisfy your requirement. The added table notes are as follows:

Several CircRNAs (e.g., CircCREBBP, CircFBXW4, CircPSD3, hsa_circ_0070963, hsa_circ_0004018, mmu_circ_34116, CircDIDO1, CircMTO1 and mmu_circ_0000623) have been demonstrated to inhibit liver fibrosis. Some CircRNAs are upregulated in liver fibrotic tissues and HSCs, which promote the progression of liver fibrosis through multiple mechanisms (e.g., acting as miRNA sponges, binding to functional miRNAs, and regulating gene expression at the transcriptional or post-transcriptional level.

  1. The abstract is not written very fluently. The functions should be better summarized and "etc." should not occur here. Also, a general overview should be given here and not a detailed description of individual CircRNAs.

Answer: Thanks for your constructive suggestion. According to your advice, we have updated the abstract section, which is as follows:

Liver fibrosis represents the reversible pathological process with the feature of the over-accumulation of extracellular matrix (ECM) proteins within the liver, which results in the deposition of fibrotic tissues and liver dysfunction. Circular noncoding RNAs (CircRNAs) have the characteristic closed loop structures, which show high resistance to exonuclease RNase, making them far more stable and recalcitrant against degradation. CircRNAs increase target gene levels by playing a role of a microRNA (miRNA) sponge. Further, they combine with proteins or play a role of RNA scaffolds or translate proteins to modulate different biological processes. Recent studies have indicated that CircRNAs play an important role in the occurrence and progression of liver fibrosis, and may be the potential diagnostic and prognostic markers for liver fibrosis. This review summarizes the CircRNAs roles and explores their underlying mechanisms, with special focus on some of the latest research into key CircRNAs related to regulating liver fibrosis. Results in this work may inspire fruitful research directions and applications of CircRNAs in the management of liver fibrosis. Additionally, our findings lay critical theoretical foundation for applying CircRNAs in diagnosing and treating liver fibrosis.

  1. The introduction should also start with a general setup and not with secondary aspects of the disease.

Answer: Thanks for your constructive suggestion. According to your advice, we have rewritten the beginning of the introduction, and changed ‘the study reviews’ into ‘review summarizes’ in the last sentence of the introduction. The details are as follows:

Various chronic irritations can damage the liver, which thus causes inflammation and necrosis of hepatocytes, activates the hepatic stellate cells, and produces the significant accumulation of extracellular matrix (ECM) that is extremely rich in type I collagen and other collagens, eventually resulting in liver fibrosis. Liver fibrosis, a common pathological change in various chronic liver diseases, presents as a precursor to liver cirrhosis, and it finally develops into liver failure and hepatocellular carcinoma (HCC).

  1. Comments on the Quality of English Language:The overall English is not sufficient, the review needs to be corrected by a native speaker. For example, it is not a “study” but a review.

Answer: Thanks for your suggestions on the language of our manuscript. According to your advice, we invited Frank Boehm to check the tenses and grammar through the WHOLE manuscript, and believe that the language is now acceptable for the review and publication process.

Frank Boehm: a Canadian English speaker, who has been working on academic manuscript editing for Lakehead University, the University of Guelph, as well as other universities in Canada and China, for more than 20 years. He has also published one book (Nanomedical Device and Systems Design Challenges, Possibilities, Visions) (2013) on the topic of nanomedicine, with three upcoming books in progress.

Reviewer 3 Report

This is a well-prepared and significant review summarizing currently reported circRNAs in the setting of liver fibrosis. Does any circRNA regulate macrophage polarization? Are there any circRNA direct binding proteins that have been reported, which are involved in the pathogenesis of liver diseases?

Author Response

Reviewer #3: This is a well-prepared and significant review summarizing currently reported CircRNAs in the setting of liver fibrosis. Does any CircRNA regulate macrophage polarization? Are there any CircRNA direct binding proteins that have been reported, which are involved in the pathogenesis of liver diseases?

  1. Does any CircRNA regulate macrophage polarization?

Answer: Thanks for your constructive suggestion. According to your advice, we searched for literatures and found that hsa_circ_0074854 promoted the progression of HCC by interacting with human antigen R (HuR, an RNA-binding protein) and inhibiting the exosome-mediated M2 polarization of macrophages after searching the literature. But whether CircRNAs regulate the polarity of macrophages in liver fibrosis has not been involved yet. With the further development of the research, there may be relevant studies on the regulation of the polarity of macrophages by CircRNA in liver fibrosis in the future. The above-mentioned contents have been added into the line 108-111 of the revised manuscript.

  1. Are there any CircRNA direct binding proteins that have been reported, which are involved in the pathogenesis of liver diseases?

Answer: Thanks for your constructive suggestion. After searching for literatures, we found that CircRNA-SORE can bind to the master oncogenic protein YBX1 in the cytoplasm, which then prevents YBX1 nuclear interaction with the E3 ubiquitin ligase PRP19 and thus blocks the PRP19-mediated YBX1 degradation, and CircRNA-SORE is involved in the pathogenesis of HCC. However, whether CircRNAs regulate liver fibrosis by directly binding to proteins has not yet been investigated, and relevant studies may be reported in the future. The above-mentioned contents have been added into the line 101-103 of the revised manuscript.

Round 2

Reviewer 1 Report

Thank you for the additional sections in the review. 

Author Response

Reviewer #1:

Comments and Suggestions for Authors

Thank you for the additional sections in the review.

Answer: Thanks for your review and approval.